# Advanced EPI-X4 Derivatives Covalently Bind Human Serum Albumin Resulting in Prolonged Plasma Stability

**DOI:** 10.3390/ijms232315029

**Published:** 2022-11-30

**Authors:** Armando Rodríguez-Alfonso, Astrid Heck, Yasser Bruno Ruiz-Blanco, Andrea Gilg, Ludger Ständker, Seah Ling Kuan, Tanja Weil, Elsa Sanchez-Garcia, Sebastian Wiese, Jan Münch, Mirja Harms

**Affiliations:** 1Core Facility Functional Peptidomics, Ulm University Medical Center, 89081 Ulm, Germany; 2Core Unit Mass Spectrometry and Proteomics, Ulm University Medical Center, 89081 Ulm, Germany; 3Max Planck Institute for Polymer Research, 55128 Mainz, Germany; 4Center of Medical Biotechnology, University of Duisburg-Essen, 47057 Duisburg, Germany; 5Institute of Molecular Virology, Ulm University Medical Center, 89081 Ulm, Germany

**Keywords:** EPI-X4, albumin carrier, peptide stability, CXCR4 antagonist

## Abstract

Advanced derivatives of the **E**ndogenous **P**eptide **I**nhibitor of C**X**CR**4** (EPI-X4) have shown therapeutic efficacy upon topical administration in animal models of asthma and dermatitis. Here, we studied the plasma stability of the EPI-X4 lead compounds WSC02 and JM#21, using mass spectrometry to monitor the chemical integrity of the peptides and a functional fluorescence-based assay to determine peptide function in a CXCR4-antibody competition assay. Although mass spectrometry revealed very rapid disappearance of both peptides in human plasma within seconds, the functional assay revealed a significantly higher half-life of 9 min for EPI-X4 WSC02 and 6 min for EPI-X4 JM#21. Further analyses demonstrated that EPI-X4 WSC02 and EPI-X4 JM#21 interact with low molecular weight plasma components and serum albumin. Albumin binding is mediated by the formation of a disulfide bridge between Cys10 in the EPI-X4 peptides and Cys34 in albumin. These covalently linked albumin–peptide complexes have a higher stability in plasma as compared with the non-bound peptides and retain the ability to bind and antagonize CXCR4. Remarkably, chemically synthesized albumin-EPI-X4 conjugates coupled by non-breakable bonds have a drastically increased plasma stability of over 2 h. Thus, covalent coupling of EPI-X4 to albumin in vitro before administration or in vivo post administration may significantly increase the pharmacokinetic properties of this new class of CXCR4 antagonists.

## 1. Introduction

The C-X-C chemokine receptor type 4 (CXCR4) is a G protein-coupled receptor expressed ubiquitously in the human body. The only known chemokine ligand for CXCR4 is CXCL12, which induces activation of several G-protein dependent and independent downstream pathways and consequently receptor internalization upon binding. The CXCR4/CXCL12 signaling axis has various functions in the human body, for example, in hematopoiesis, lymphocyte trafficking, and organ development [1]. However, dysregulation of the receptor-ligand pair leads to pathological processes. CXCR4 and CXCL12 are both overexpressed in several diseases, including inflammatory disorders and cancer [1,2]. Therefore, targeting the CXCR4/CXCL12 axis provides a strategy for the therapeutic treatment of such diseases [1,3]. However, the only approved CXCR4 antagonist so far is the small molecule AMD3100 (Plerixafor). Due to several side effects, its application is, however, limited to autologous stem cell transplantation in some forms of cancer [4].

**E**ndogenous **P**eptide **I**nhibitor of C**X**CR**4** (**EPI-X4**) is a 16-mer fragment of human serum albumin (HSA 408-423) generated by aspartic proteases under acidic conditions [5,6,7]. The peptide has originally been identified as an inhibitor for CXCR4-tropic HIV-1 [7], as it binds to CXCR4, thereby abrogating the interaction of the HIV-1 gp120 glycoprotein with the receptor. Moreover, EPI-X4 antagonizes CXCL12-dependent signaling and migration of cancer and immune cells and serves as a CXCR4 inverse agonist as the peptide reduces basal G-protein signaling in the absence of the chemokine [7]. We recently performed structure–activity relationship studies to optimize the antagonistic activity of EPI-X4 for therapeutic use [7,8,9]. The new lead derivatives EPI-X4 WSC02 (IVRWSKKVP**C**VS) and EPI-X4 JM#21 (ILRWSRKLP**C**VS) (Table 1, Appendix A) are 12 amino acids long derivatives that are active at nM concentrations and have shown therapeutic activities in inflammatory [8] and cancer mouse models [10]. These peptides both bind to the minor binding pocket of CXCR4, mainly by interaction of the first seven amino acid residues [11]. In contrast, the C-terminus is hardly involved in receptor binding and protrudes from the pocket with conformational flexibility. Thus, functional groups (e.g., fluorophores, polymers, and chelators) can be coupled to the peptides C-terminus without interrupting receptor interaction [12]. To do so, a cysteine was introduced in the sequence of both peptides at position 10 [7,8].

A fundamental requirement for successful drug development is the characterization of compound stability in human plasma. Especially peptide drugs are well known for their rapid degradation, as this body fluid is rich in proteases, leading to fast inactivation [13]. The stability of therapeutic agents is usually determined by mass spectrometry (MS)- based methods, which directly quantify the presence of the intact analyte [14,15]. However, these methods are focused on the molecular integrity of the analyte, which may only be partially related to its activity as some analyte fragments may still be active. Plasma is a complex mixture of proteins and other constituents, so additional treatment may be necessary to reduce the interference of plasma components due to their high concentration or interaction with the analyte. For example, an adequate precipitation protocol to reduce the same complexity while maintaining the analyte in solution may be necessary before MS detection. In addition, as observed in the present work, additional agents are required to break the interaction of the analyte with those plasma components that hinder the detection. Therefore, we have recently developed a method for stability evaluation of CXCR4 ligands in body fluids, which is based on the remaining competition of a CXCR4 inhibitor with a CXCR4 antibody for receptor binding [16]. In contrast to the MS-based method, this assay allows one to find changes in the affinity for a receptor over time independently of chemical modification. It is also able to detect protein-bound drug fractions if still active [16].

Here, we set out to determine the stability of the advanced EPI-X4 derivatives WSC02 and JM#21 in human plasma using both assays, the chromatography/mass spectrometry-based approach for detecting free peptide and the antibody-competition assay that determines functional activity. We unexpectedly found that both assays resulted in seemingly contradictory results, as both peptides showed a 4–10-fold longer half-live in the functional assay. We analyzed the underlying mechanism and found that the cysteine at position 10 in both EPI-X4 derivatives formed a disulfide bridge with the Cys34 in serum albumin. The covalently linked HSA-EPI-X4 conjugate could not be detected by the applied MS protocol but interestingly retained CXCR4-binding activity. We also showed that the formation of HSA-EPI-X4 conjugates resulted in increased proteolytic stability as compared with the free peptide. Thus, the formation of stable HSA-peptide conjugates in vivo may result in improved pharmacokinetic properties of EPI-X4 derivatives WSC02 and JM#21, supporting further (pre)clinical development.

## 2. Results

To determine the plasma stability of the optimized EPI-X4 derivatives EPI-X4 WSC02 (Table 1, Appendix A) and EPI-X4 JM#21 (Table 1, Appendix A), first, we used a recently established assay that is based on the remaining function of the peptides to compete with a CXCR4 antibody for receptor interaction [16]. The half-life values in human plasma estimated from this functional assay were 7 min for JM#21, and 9 min for WSC02, (Figure 1a,b), in agreement with previous results [8,16]. To confirm data obtained by the antibody-competition assay, we also assessed the stability of the peptides by mass spectrometry. Surprisingly, the amount of both peptides decreased rapidly, yielding much shorter half-life values of 1.7 min for JM#21 and only 0.7 min for WSC02 (Figure 1a,b). This discrepancy between the functional and mass spectrometry assay could be explained by the different parameters measured in the experiments. On the one hand, the functional assay determines the biological activity of the peptide which originates from the peptide itself and also from products possibly formed during the incubation. On the other hand, the mass spectrometry assay measures the amount of the original peptide during incubation, which could be affected by chemical modification or non-covalent interactions with other molecules, in addition to the action of proteases present in plasma.

EPI-X4 JM#21 and WSC02 have a free cysteine (Cys) residue at position 10, which could be targeted at physiological pH by oxidizing agents. We tested whether the peptides could interact reversibly in plasma via disulfide bridges or non-covalent forces. For this purpose, aliquots were separated from the peptide-spiked plasma samples at t = 10 min (after a considerable decrease of peptide signal intensity measured by MS) and treated either with 0.1% TFA as a reference, 10 mM DTT (reducing agent to break disulfide bonds), 6M GuHCl (chaotropic agent to break non-covalent interactions with other components), or a mixture of DTT and GuHCl (Figure 2a) to combine the effects of these agents.

The samples diluted with 0.1% TFA yielded very weak MS signals (Figure 2a). Treatment with GuHCl alone released only a small amount of peptide, which was detectable by MS (Figure 2a). In contrast, DTT treatment led to a release of the peptides indicating covalent thiol interactions between the peptide and other proteins. The combination of GuHCl + DTT released a much higher amount, suggesting a mixed-mode interaction of the peptide with plasma components. Such interactions seem to imply the formation of disulfide bridges broken in the presence of DTT, and the non-covalent interaction disrupted by the denaturant reagent GuHCl, probably mediated by the interaction of GuHCl with polar residues, altering the electrostatic interactions among them [17,18].

To gain further insight into the interaction of JM#21 and WSC02 with plasma components such as human serum albumin (HSA), we performed a size-exclusion HPLC fractionation of plasma spiked with the peptides. All fractions were collected and treated with DTT+ GuHCl to release the peptides and enhance their further detection by mass spectrometry (Figure 2b,c). Two groups of fractions containing WSC02 or JM#21, respectively, were detected: one group had the same retention time as HSA, which was used as standard (red bars vs. green dotted traces in Figure 2b,c), and the other group partially matched the retention time of the peptides (red bars vs. purple dash-dotted traces in Figure 2b,c). HSA represents the largest thiol pool in human plasma due to its high abundance (35–45 mg/mL, approximately 0.6 mM) and the presence of a free sulfhydryl group in Cys34. Our results indicate an interaction with albumin which is mediated by the formation of a disulfide bridge with Cys34.

Aiming to demonstrate the interaction of the peptides with Cys34 of HSA, a sample of plasma spiked with JM#21 or WSC02 was digested and analyzed by LC-MSMS. The proteolytic fragment expected to contain Cys34 is ALVLIAFAQYLQQCPFEDHVK (residues to 21-41 of the mature chain). Given that JM#21 and WSC02 are also cleaved by trypsin, the expected fragment of albumin should be bound to a fragment of JM#21 (LPCVS) or WSC02 (VPCVS) by a disulfide bond. A fragment of 3100.753 Da (monoisotopic mass), comprising albumin 21-41 linked to KLPCVS (+643.33 Da, one missing cleavage in JM#21) was found (Appendix A). In the case of WSC02, a fragment of 2933.484 Da was found, comprising albumin 21-41 linked to VPCVS (+501.23 Da) (Appendix A). None of these two fragments were found after reduction + carbamidomethylation, which yielded the carbamidomethylated albumin fragment, ALVLIAFAQYLQQC(+57.02)PFEDHVK of 2489.2799 Da (Appendix A).

As observed in the size-exclusion HPLC experiment, JM#21 and WSC02 were also detected in the low molecular weight region of the chromatogram, indicating that a population of these peptides is either free or interacting with low molecular weight compounds via disulfide bonds or non-covalent forces. Several low molecular weight reducing agents are present in plasma, such as cysteine, cysteinylglycine, glutathione, and homocysteine, although at a much lower concentration (12–20 µM of reduced thiol in total) than albumin (0.6 mM, 75% of reduced thiol) [19]. Figure 3a,b show a diversity of compounds formed by disulfide bridge formation between the peptides JM#21 and WSC02 with some of these Cys-containing compounds, as well as their dimers. Since cysteine residues are susceptible to the action of oxidant agents, some oxidation products (+31.98 Da and +47.94 Da) were also detected. The analysis was performed for aliquots at several time points to study the change in concentration over time, and thus, gain insights into the influence of these molecules on the stability of JM#21 and WSC02 in plasma. In both cases, the formation of a disulfide bond with cysteine is the most notable fact, especially for WSC02. These data are in concordance with the results from gel filtration (Figure 2b,c), which show a higher ratio of JM#21(albumin)/JM#21(LMW) than WSC02, which shows a higher amount in the low molecular weight region. It is also interesting that Cys-bound JM#21 and WSC02 seem to be stable over time, probably contributing to the stability of these compounds in plasma.

To study the influence of the thiol reaction between the peptides and plasma proteins, next, we designed derivatives in which the Cys residues were replaced by serine (Ser) (WSC02/JM#21 C10S) or modified by carbamidomethylation (WSC02/JM#21 Cdm). Themodifications both had no influence on peptide activity, as all variants were still able to compete with the CXCR4 antibody for receptor binding (Figure 4a). All variants were then spiked into human plasma and treated with either TFA, DTT, GuHCl, or a combination of DTT and GuHCl, as described above. In contrast to the unmodified peptides, the modified variants were detectable after dilution with 0.1% TFA without treatment with a reducing (DTT) or chaotropic agent (GuHCl). In addition, the amounts of detectable peptides remained unchanged after treatment with DTT alone, as expected (Figure 4b,c). In contrast, adding GuHCl led to a strong release of all variants, further highlighting the role of electrostatic interactions for EPI-X4 derivatives in plasma.

To analyze if the thiol reaction between the Cys-containing peptides and plasma proteins might impact their resistance to enzymatic degradation, WSC02, JM#21 as well as the non-reactive variants were tested for their stability in human plasma. Stability of the peptides was measured by both the functional assay and mass spectrometry, as described above. As expected, the Cys-containing peptides WSC02 and JM#21 were rapidly undetectable by MS (0.1% TFA treatment). The addition of GuHCl + DTT before the MS analysis led to the release of the intact peptides from plasma proteins, as observed above (Figure 2a), revealing a “real” half-live of 6 min (WSC02) and 5 min (JM#21), confirming previous studies [8,16]. In contrast, peptides without a reactive Cys-thiol (C10S or Cdm) showed lower half-lives of 4 min and 5 min (WSC02) or 4 min and 3 min (JM#21) (Figure 5a–d). Similar results were obtained by the functional stability assay, which was based on the remaining activity of the peptide-plasma mixture to compete with a CXCR4 antibody [16]. Whereas for the Cys-containing peptides a half-life of about 9 min (WSC02) and 6 min (JM#21) was determined, the modified variants revealed lower half-lives of 6 min and 7 min for WSC02 Cdm and Ser, respectively, and 3 min and 4 min for JM#21 Cdm and Ser, respectively (Figure 5e,g). For the WSC02 variants, activity was reduced by 93% (Cdm) and 92% (Ser) after 30 min, whereas the Cys-containing WSC02 lost only 23% of its activity (Figure 5f). Similarly, the activity of JM#21 variants was decreased by 95% (Cdm) and 91% (Ser) after already 15 min, whereas the Cys-containing JM#21 lost only 69% of its activity (Figure 5h). For both, WSC02 and JM#21 variants, activity was completely lost after 60 min and 30 min, respectively. In contrast, Cys-containing variants remained 20% (WSC02) and 10% (JM#21) active. (Figure 5f,h), confirming results obtained by LC-MS. Thus, Cys-containing EPI-X4 variants have significantly higher stability in human plasma than their counterparts without reactive thiol groups.

Plasma protein-bound EPI-X4 derivatives are not detectable in LC-MS without DTT + GuHCl treatment; however, they are readily detectable in functional assays. This suggests that EPI-X4 derivatives are able to bind to CXCR4 while their cysteine residues are covalently attached to larger proteins, such as HSA. Accordingly, we used computational modeling to investigate whether such conjugates could sterically attach to the binding pocket of the CXCR4 receptor while allowing the peptide to establish all relevant interactions with the protein, as previously established for EPI-X4 JM#21 peptide [11]. To this end, we built a model of the CXCR4–JM#21_HSA conjugate by linking human serum albumin (PDB ID 1AO6 [20]) via the free thiol group of Cys34 to the structure of the CXCR4–JM#21 complex, previously reported by us [11]. Then, the structure was minimized and subjected to short relaxation dynamics to correct possible clashes and bad contacts (see computational details). The resulting model structure (Figure 6) indicates that HSA has the possibility of joining JM#21 via an S-S bridge. The HSA_JM#21 construct can bind CXCR4 in the binding mode previously reported for JM#21, i.e., without causing steric clashes that irremediably disrupt the known binding between JM#21 and CXCR4.

To prove that the peptide conjugation via the unpaired cysteine is the reason for the increased stability of the Cys-containing EPI-X4 derivatives, we prepared covalent HSA-EPI-X4 protein–peptide conjugates. For this purpose, a bifunctional linker is required to attach the EPI-X4 variants to the protein. Since Cys residues are present in the structures of the optimized EPI-X4 variants JM#21 and WSC02, and HSA contains an unpaired Cys at position 34, we used a maleimide-bis-sulfone reagent (Figure A1) for the conjugation [21]. The reagent was selected to achieve a stepwise synthesis controlled by pH to prevent the formation of homodimeric EPI-X4 variants (Figure A1). First, compound **1** was reacted with a peptide variant (WSC02 or JM#21) at pH 6 through a maleimide-thiol reaction. The resultant product, bis-sulfone peptide, was isolated in 59% and 21% yield for WSCO2 and JM#21, respectively (Figure A2 and Figure A3). For the second conjugation step to HSA, the bis-sulfone peptide was incubated at pH 8 overnight and underwent an elimination in situ to obtain the “reactive” mono-sulfone peptide. The mono-sulfone peptide was then applied to a solution of HSA (Figure A4). The resultant HSA-EPI-X4 conjugates (HSA-WSC02 and HSA-JM#21) were obtained through the reaction of the mono-sulfone with unpaired cysteine on HSA with 66% and 85% yields, respectively. The degree of labeling was determined by fluorescent thiol quantification, and the conjugates were characterized by MALDI-TOF MS, yielding a molecular mass of 68 kDa, corresponding to a single EPI-X4 peptide attached to albumin (Figure A5).

Thereafter, the activity of both HSA-EPI-X4 conjugates was investigated to study the influence of the conjugation. The conjugates both remained active in the antibody competition assay, although with decreased activity, showing that both peptide-HSA conjugates still bind to the CXCR4 binding pocket (Appendix A). Then, we incubated both conjugates in human plasma and compared their remaining activity to unconjugated EPI-X4 WSC02 and JM#21 in the antibody competition assay. As shown in Figure 7a,c, both peptides and the respective conjugates compete with the CXCR4 antibody for receptor interaction (black line). However, activity decreased over time for the unconjugated variants, resulting in a shift of competition curves and decreasing activities with half-lives <30 min. In contrast, activity remained constant over time for JM#21 as well as for WSC02 conjugates (Figure 7a–d). This confirms that for EPI-X4 variants, covalent attachment to endogenous serum albumin via thiol groups leads to increased resistance against degradation in plasma; thus, it might be a promising approach for further drug development.

## 3. Discussion

Here, we show that advanced derivatives of the CXCR4 antagonist EPI-X4 form covalent interactions with serum proteins, particularly albumin, which increase proteolytic resistance and stability in plasma. The interaction is mediated by the formation of a disulfide bridge between Cys10 in the peptides and Cys34 in albumin. Remarkably, both albumin-bound peptides retain their ability to bind and antagonize CXCR4 expressed at the cell surface, suggesting that albumin-EPI-X4 conjugates have excellent prospects for further (pre)clinical development as systemically applied drugs.

EPI-X4 is a linear peptide derived from human serum albumin (HSA), which specifically binds to CXCR4, thereby, acting as an antagonist and inverse agonist [5,6,7]. To increase the CXCR4 binding affinity and to clarify the mechanism of CXCR4 inhibition, we performed SAR studies that resulted in the development of truncated and sequence optimized derivatives EPI-X4 WSC02 (IVRWSKKVPCVS) and EPI-X4 JM#21 (ILRWSRKLPCVS) [7,8], which have an about 100-fold and 300-fold higher antagonistic activity as compared with EPI-X4, respectively [7,8]. In addition, both peptides contain a cysteine (Cys) residue near the C-terminus, which was introduced for the conjugation of functional groups and polymers [12]. Using the CXCR4 antibody-competition assay, we previously found and here confirmed that both derivatives are inactivated in human plasma, with half-lives of only 9 min for EPI-X4 WSC02 and 6 min for EPI-X4 JM#21 [8,16]. However, in contrast to this functional assay, both peptides were readily undetectably when MS was used for detection (sample dilution with 0.1% TFA), and only appeared after treatment of samples with DTT (disrupting disulfide bridges) and GuHCl (chaotropic agent). These facts suggest an interaction with thiol-containing plasma components, such as HSA.

HSA is a 585 amino acid residues long protein with concentrations ranging from 35 to 50 g/L in plasma. HSA possesses several essential functions in our body, such as osmotic pressure maintenance, and transportation of fatty acids, ions, and waste products [22]. In addition, HSA has an extraordinarily long in vivo circulation half-life of about three weeks due to its interaction with the neonatal Fc receptor (FcRn), broadly expressed throughout the body, rescuing its ligands from cellular degradation [23]. HSA carries 17 disulfide bridges and a single free Cys residue (Cys34). Cys34 is the most abundant thiol in human plasma. In healthy individuals, about 70% of HSA Cys34 exists in its reduced form, representing the strongest reducing power in this body fluid [24].

Considering the reactivity of this particular residue and the high concentration of the free thiol group (Cys34) in HSA, we suspected EPI-X4 WSC02 and JM#21 to interact primarily with albumin via their respective Cys residues by forming a disulfide bridge with Cys34. Nonetheless, the interaction with albumin seems to be mediated not only by the formation of a disulfide bridge since a reducing agent (DTT) was not sufficient to fully deliver the peptide, and only a combined treatment with a chaotropic agent (GuHCl) allowed the massive release and detection of the peptides. As expected, when the WSC02 and JM#21 thiol-lacking versions (Cys changed for Ser, Cys carbamidomethylated (Cdm)) were exposed to plasma, DTT had only a minor influence on the detection of the peptides, whereas GuHCl led to a massive release, indicating the mediation of non-covalent forces in the interaction with plasma components. Even without GuHCl, WSC02/JM#21-Ser and WSC02/JM#21-Cdm were easy to detect after diluting with 0.1% TFA only. In both cases, this could be due to the presence of a peptide population weakly (or not) interacting with plasma components so that these peptides can be detected under either condition. In addition, low pH may induce strong intramolecular electrostatic repulsion and partial denaturation in proteins such as albumin, weakening the interaction with the peptides (especially if mediated by electrostatic forces) and contributing to their release [25]. However, their respective half-lives were shorter than the half-lives of the Cys-containing peptides after being treated with GuHCl+DTT, indicating that the Cys residue is relevant for the stability of the peptides by forming a disulfide bridge. Additionally, we observed a significant residual amount of (Cys-containing) WSC02 and JM#21 even after 100 min, which is not present in WSC02/JM#21-Ser or WSC02/JM#21-Cdm. Apparently, a population of these peptides is protected against the action of proteases in plasma, which is another indicator of the mixed-mode interaction with plasma components such as albumin; this can also imply the interaction with more than one molecule.

Additionally, interaction with plasma low molecular weight compounds having free Cys residues is feasible despite the low concentration (<20 µM) of free thiol groups from these molecules as compared with the fraction of active thiol from albumin (around 0.4 mM). In the future, it would be interesting to test the activity and stability of synthetic versions of these molecules (or fragments) bound to WSC02 and JM#21 by disulfide bonds. Specifically, Cys binding to WSC02 and JM#21 seems to confer some protection from enzymatic degradation, which was shown by a significant residual amount of WSC02-Cys and JM#21-Cys even after one hour of incubation in plasma. Thus, besides albumin, free cysteine in plasma could play an additional stabilizing role by forming disulfide bridges with WSC02 and JM#21.

In vivo drug molecules are often bound to plasma proteins, such as serum albumin [26]. In fact, plasma protein binding is often associated with increased stability and longer circulation half-life of therapeutic agents. However, increased stability most often comes with a cost of activity, as for the majority of drugs only the unbound fraction is able to bind to its target [26]. Here, we show that Cys-containing EPI-X4 derivatives gain stability upon interaction with HSA, and also retain their ability to interact with CXCR4 (Figure 5). This property is most likely enabled because EPI-X4 and its derivatives interact with the CXCR4 binding pocket mainly by the first seven N-terminal amino acids, while the peptide C-terminus protrudes from the pocket [11]. Thus, large attachments at the C-terminus (such as HSA bound to EPI-X4-Cys) still allow peptide binding to the receptor, as shown by computational docking analyses (Figure 6), and experimentally (Figure 7). Moreover, CXCR4 is expressed at the cell surface, where it is directly accessible for EPI-X4-albumin conjugates.

Due to its extraordinarily long half-life and high abundance, HSA became a popular drug carrier over the past years. One strategy to improve the stability and circulation half-life of drugs is the conjugation to thiol-reactive linkers, such as maleimide or sulfone linkers, targeting HSA-Cys34. Clinically, one of the most advanced maleimide-functionalized peptides is aldoxorubicin, a doxorubicin derivative successfully tested in a clinical phase III trial as a cancer chemotherapeutic agent [27]. As compared with its precursor, aldoxorubicin is covalently attached to albumin in circulation and shows mitigated cardiac toxicity attributed to purposive tumor targeting [28]. Another attractive maleimide-conjugated clinical candidate is CJC-1131, an analogue of GLP-1. CJC-1131 has been shown to be extremely stable in vivo, with half-lives of 15 to 20 h in rats and more than ten days in humans after subcutaneous injection [29,30]. Here, we additionally show that EPI-X4 derivatives covalently bound to human serum albumin via a sulfone linker are still active and show a highly increased resistance against plasma enzymes. It is conceivable that these EPI-X4-HSA conjugates may exhibit a long-lasting activity in vivo leading to increased therapeutic efficacies. It would be highly interesting to study whether the design of EPI-X4 derivatives containing thiol-reactive linkers may allow albumin binding after systemic administration of the peptide. The circulating albumin preferentially accumulates in tumors [31] or inflamed tissues [32,33], delivering optimized EPI-X4 molecules to the sites where they are therapeutically required.

## 4. Materials and Methods

### 4.1. Peptide Synthesis

Peptides were synthesized automatically on a 0.10 mmol scale using standard Fmoc solid phase peptide synthesis techniques with the microwave synthesizer (Liberty blue, CEM). Amino acids were obtained from Novabiochem (Merck KGaA, Darmstadt, Germany). Peptides were purified using reverse phase preparative high-performance liquid chromatography (HPLC, Waters) in an TFA/acetonitrile/water gradient under acidic conditions on a Phenomenex C18 Luna column, and then lyophilized on a freeze dryer (Labconco). Prior to use, peptides were diluted in PBS at a stock concentration of 3 mM.

### 4.2. Carbamidomethylation of the Cys Residue in JM#21 and WSC02

Peptides were reduced with 5 mM DTT + 50 mM NH_4_HCO_3_ for 20 min at room temperature, and then carbamidomethylated with 50 mM iodoacetamide for 20 min at 37 °C. The quenching step was performed with 10 mM DTT.

### 4.3. Half-Life Determination by Mass Spectrometry

A 50 µL-aliquot of human plasma was spiked with 7 µM JM#21 or WSC02. The mixture was stirred for a few seconds and incubated at 37 °C. Five microliter aliquots were separated after 0, 5, 10, 20, 30, 45, 60, 90, and 120 min. Aliquots were immediately diluted with 1.5 mL 0.1% TFA and stored at −80 °C. The experiment was performed in triplicate. Additionally, for the measurement of denatured samples prior to MS measurement, all of the samples stored in 0.1% TFA were ten-fold diluted with 50 mM NH_4_HCO_3_ + 10 mM DTT + 6M GuHCl. A 15 uL-aliquot of every sample was used for mass spectrometry analysis as follows: samples were measured using an Orbitrap Elite Hybrid mass spectrometry system (Thermo Fisher Scientific, Bremen, Germany) online coupled to an U3000 RSLCnano (Thermo Fisher Scientific, Idstein, Germany) employing an Acclaim PepMap analytical column (75 μm × 500 mm, 2 μm, 100 Å, Thermo Fisher Scientific, Bremen, Germany) at a flow rate of 250 nL/min. Using a C18 μ-precolumn (0.3 mm × 5 mm, PepMap, Dionex LC Packings, Thermo Fisher Scientific, Bremen, Germany), samples were preconcentrated and washed with 0.1% TFA for 5 min at a flow rate of 30 μL/min. The subsequent separation was carried out using a binary solvent gradient consisting of solvent A (0.1% FA) and solvent B (86% ACN, 0.1% FA). The column was initially equilibrated in 5% B. In the first elution step, the percentage of B was raised from 5 to 15% in 5 min, followed by an increase from 15 to 40% B in 30 min. The column was washed with 95% B for 4 min and re-equilibrated with 5% B for 19 min. The mass spectrometer was equipped with a nanoelectrospray ion source and distal-coated SilicaTips (FS360-20-10-D, New Objective, Woburn, MA, USA). The instrument was externally calibrated using standard compounds (LTQ Velos ESI Positive Ion Calibration Solution, Pierce, Thermo Scientific, Rockford, IL, USA). The system was operated using the following parameters: spray voltage, 1.5 kV; capillary temperature, 250 °C; S-lens RF level, 68.9%. XCalibur 2.2 SP1.48 (Thermo Fisher Scientific, Bremen, Germany) was used for data-dependent tandem mass spectrometry (MS/MS) analyses. Full scans ranging from m/z 370 to 1700 were acquired in the Orbitrap at a resolution of 30,000 (at m/z 400) with automatic gain control (AGC) enabled and set to 106 ions and a maximum fill time of 500 ms. Up to 20 multiply-charged peptide ions were selected from each survey scan for collision-induced fragmentation (CID) in the linear ion trap using AGC set to 10,000 ions and a maximum fill time of 100 ms. For MS/MS fragmentation, normalized collision energy of 35% with an activation q of 0.25 and an activation time of 30 ms was used.

For visualization in the XCalibur Qual Browser 2.0 (Thermo Fisher Scientific, Bremen, Germany), extracted ion xhromatograms (XICs) of the most intense isotope (±20 ppm), as theoretically predicted, at z = 3 or z = 4 were calculated. Using default parameters within the QualBrowser, peak areas were calculated and exported. Half-lives were determined by fitting the data (peak area vs. time) to a one-phase exponential decay model using GraphPad Prism 8.4.3 (GraphPad Prism Software LLC, San Diego, CA, USA).

### 4.4. Influence of Reducing and Chaotropic Agents on Peptide Detection by Mass Spectrometry

The influence of several reagents on peptide detection was tested with aliquots separated (peptide + plasma, 300-fold diluted in 0.1%TFA) at t = 10 min. The aliquots were analyzed by LC-MS/MS, as previously described, after 10-fold dilution in 0.1% TFA, 6M GuHCl, 50 mM NH_4_HCO_3_ + 10 mM DTT, and 50 mM NH_4_HCO_3_ + 10 mM DTT + 6M GuHCl, respectively.

### 4.5. Size-Exclusion Fractionation of Peptides in Plasma

A 5 µL aliquot of the peptides in plasma, after incubation at 37 °C for 10 min, was injected into a size-exclusion column Yarra SEC-2000 (Phenomenex, CA, USA) of dimensions 7.8 × 300 mm, 3 µm particle size. The separation was done at a flow rate of 0.5 mL/min using a buffer composed of 25 mM NaH_2_PO_4_ + 25 mM Na_2_HPO_4_ + 250 mM KCl (pH 6.65). Fractions were collected every one minute, and then analyzed by mass spectrometry in denaturing conditions (1/10 in 50 mM NH_4_HCO_3_ + 10 mM DTT + 6M GuHCl) as previously described. HSA, WSC02, and JM#21 were used as references for comparing with the chromatographic profiles of peptide in plasma.

### 4.6. Digestion of Plasma Spiked with WSC02 or JM#21

A 5 µL aliquot of plasma spiked with WSC02 or JM#21 was mixed with trypsin (50:1) and incubated at 37 degrees for 16 h. Half of the digested sample was reduced and carbamidomethylated, as described above. The digested samples, both unmodified and carbamidomethylated, were analyzed by mass spectrometry, as previously described in this section.

### 4.7. Cell Culture and Human Plasma

SupT1 cells were cultured in RPMI supplemented with 10% FCS, 100 units/mL penicillin, 100 µg/mL streptomycin, 2 mM L-glutamine, and 1 mM HEPES (Gibco). Blood was collected from healthy donors (male and female, average age 28 years) using S-Monovettes K3 EDTA (Sarstedt). Plasma was obtained by centrifugation of whole blood at 2.500× *g* for 15 min. Plasma of 6 donors was pooled and stored at −80 °C.

### 4.8. Antibody Competition

Competition of compounds with an APC-labeled antibody specific for ECL2 of CXCR4 (clone 12G5) was performed in SupT1 cells as described [16]. For this, cells (without buffer) were precooled at 4 °C and a serial dilution of compounds in PBS together with a constant concentration of antibody diluted in buffer (PBS with 1% FCS) was added. After 2 h at 4 °C, unbound antibody was removed and cells analyzed in flow cytometry (CytoFLEX, Beckman Coulter). Mean fluorescence signals (MFI) in the presence of PBS only was set 100%. Isotype control was set 0%. Half-maximal inhibitory concentrations (IC_50_) were calculated in GraphPad Prism by nonlinear regression.

### 4.9. Stability Measurements by Functional Assay

Decrease in CXCR4-binding activity after incubation of compounds in human plasma was determined to calculate functional stability as described (Harms et al. 2020). For this purpose, the peptides were dissolved at a concentration of 20 µM in 100% plasma. The first aliquot (t = 0) was immediately taken and stored at −80 °C. Aliquots were taken at given time points and also stored. To determine CXCR4 binding, all aliquots were thawed together and serially diluted in ice-cold PBS and subsequently, antibody competition was performed as described before. IC_50_ at t = 0 defined as 100% activity and decrease in activity was calculated by IC_50_ (t)**/**IC_50_ (t = 0) × 100. Half-lives were determined by one-phase decay using GraphPad Prism.

### 4.10. Peptide-HSA Conjugates

#### 4.10.1. (i) Synthesis of HSA-WSC02

First, 5 mg (2.5 µmol, 10 equiv.) of bis-sulfone-WSC02 was dissolved in 5 mL of phosphate buffer (50 mM phosphate, pH 8) and 5 mL ACN solution. The mixture was incubated for 4 h at RT to generate monosulfone-WSC02 in situ. Subsequently, 16.5 mg of HSA (0.25 µmol, 1 equiv.) were added and reaction mixture was slowly rotated over night at RT. The product was purified by Sephadex G-25 size exclusion chromatography (GE Healthcare, Chicago, IL, USA) to remove unreacted peptide. Lyophilization obtained 10.8 mg (0.16 µmol, yield 64 %) of HSA-WSC02. Degree of labeling (DOL) was quantified by using a thiol fluorescence detection kit (Invitrogen, Thermo Fischer, Waltham, MA, USA) with n-acetylcysteine as standard, the absence of thiol is correlating with the modification of cysteine-34 and obtained a DOL of 66 %. MALDI-TOF-MS (+, sinapinic acid matrix): *m*/*z*: [M + H]^+^ = 68 kDa.

#### 4.10.2. (ii) Synthesis of HSA-JM#21

First, 3 mg (1.44 µmol, 10 equiv.) of bis-sulfone-JM#21 was dissolved in 3.5 mL of phosphate buffer (50 mM phosphate, pH 8) and 2 mL ACN solution. The mixture was incubated for 3 h at RT to generate monosulfone-JM#21 in situ. Subsequently, 9.5 mg of HSA (0.14 µmol, 1 equiv.) in 4 mL of phosphate buffer (50 mM phosphate, pH 8) were added, and the reaction mixture was slowly incubated over 72 h at 4 °C. The product was purified using Sephadex G-25 size exclusion chromatography (GE Healthcare, Chicago, IL, USA) to remove unreacted peptide and after lyophilization 8.4 mg (0.12 µmol, yield: 86%) of HSA-JM#21 was obtained. Degree of labeling (DOL) was quantified by using a thiol fluorescence detection kit (Invitrogen, Thermo Fischer, Waltham, MA, USA), with n-acetylcysteine as standard, the absence of thiol is correlating with the modification of cysteine-34 and obtained a DOL of 100%. MALDI-TOF-MS (+, sinapinic acid matrix): *m*/*z*: [M + H]^+^ = 68 kDa.

### 4.11. Computational Modeling

The initial geometry of the CXCR4–JM#21_HSA complex was obtained by manually joining albumin (PDB ID 1AO6 [20]) to our previously reported model of CXCR4–JM#21 [11]. The topology of the fused molecule contains an additional disulfide bridge connecting Cys34(HSA) and Cys10(JM#21). The coordinates of the CXCR4 receptor, JM#21, and the membrane segment were initially kept as in the original model [11]. The system was solvated with TIP3P waters [34], and 11 sodium cations were added to neutralize the system’s charges. Subsequently, the whole system was minimized and relaxed with a short NVT molecular dynamics simulation (400 ps) at 300 K using NAMD 2.13 [35] and the CHARMM36m force field [36]. During the simulation, the backbone heavy atoms of JM#21, and those of the CXCR4 residues within 4 Å of JM#21 were fixed. The fixed atoms were still used for computing the forces in the system. PME was employed to treat long-range electrostatic interactions [37].

## 5. Conclusions

The CXCR4 binding peptides EPI-X4 WSC02 and EPI-X4 JM#21 interact with human serum albumin (HSA) via electrostatic forces, but also by forming a disulfide bridge with a free thiol group (Cys34) on HSA. This covalent interaction increases the stability of both peptides in human plasma, and at the same time, still sterically allows them to interact with CXCR4. Thus, linking optimized EPI-X4 derivatives to HSA via Cys34 is a promising approach for further development of drugs for the treatment of inflammatory diseases and cancer.

## Figures and Tables

**Figure 1 ijms-23-15029-f001:**
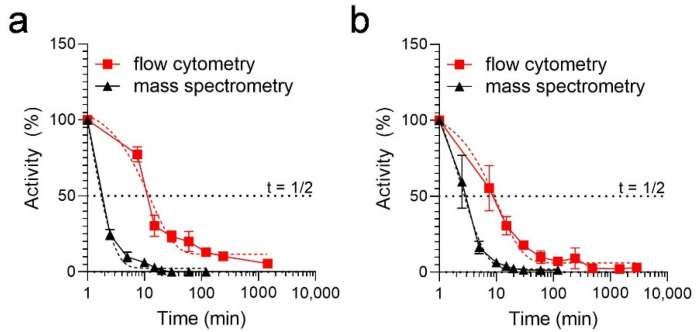
Half-lives of EPI-X4 WSC02 and EPI-X4 JM#21 in human plasma measured by a flow cytometry-based functional assay and mass spectrometry. For the functional assay, WSC02 (**a**) or JM#21 (**b**) were spiked into human plasma and incubated at 37 °C. At indicated time points, aliquots were taken and serially diluted in cold PBS. Then, plasma/peptide dilutions were added to CXCR4 expressing SupT1 cells together with a constant concentration of APC-conjugate ECL2-CXCR4 specific antibody 12G5. After 2 h, unbound antibody was removed, and cells were analyzed by flow cytometry. Remaining activity was determined described before [16]. For detection by mass spectrometry, WSC02 (a) or JM#21 (b) were spiked into plasma and incubated at 37 °C. and analyzed in LC-MS/MS. Half-lives were calculated using a one-phase decay model in GraphPad Prism. Curve fits are shown as dashed lines. Data shown are derived from 3 individual rounds of plasma incubation ± SEM.

**Figure 2 ijms-23-15029-f002:**
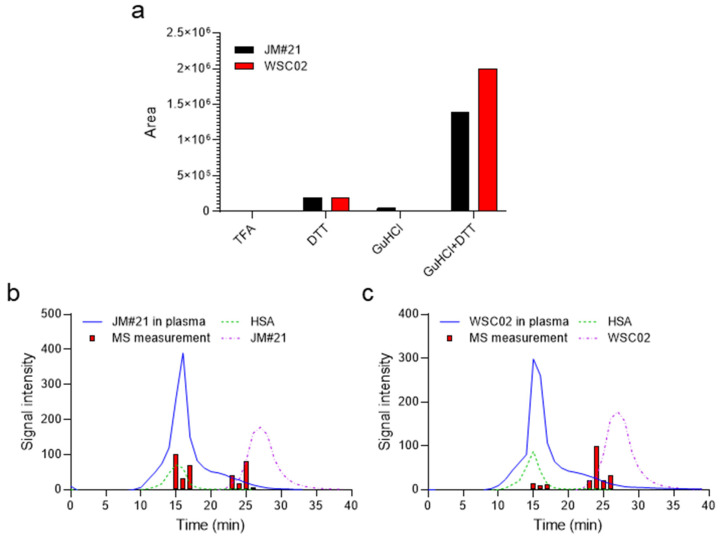
EPI-X4 JM#21 and WSC02 interact with serum albumin in human plasma via formation of thiol bonds: (**a**) Aliquots taken at t = 10 min from the stability experiments (Figure 1 a,b) of WSC02 or JM#21 were treated with either 0.1% TFA, DTT, GuHCl, or a combination of GuHCl and DTT. Afterwards samples were analyzed by mass spectrometry for the presence of free peptide. A mixture of JM#21 (**b**) or WSC02 (**c**) with human plasma was fractionated by size-exclusion HPLC (blue solid trace) and analyzed for free peptides (red bars). Size-exclusion chromatographic profiles of albumin (green dotted trace) and WSC02/JM#21 (purple dash-dotted trace) were used as references for comparison with the chromatographic profiles of the peptides in plasma. In (**b**) and (**c**), signal intensity refers to the UV absorption at 214 nm for the chromatographic profiles of the peptides (dash-dotted trace), human serum albumin (green dotted trace), and the peptides in plasma (solid blue trace), whereas it represents the normalized (0–100%) intensity of the peptides detected after mass spectrometry analysis (red bars).

**Figure 3 ijms-23-15029-f003:**
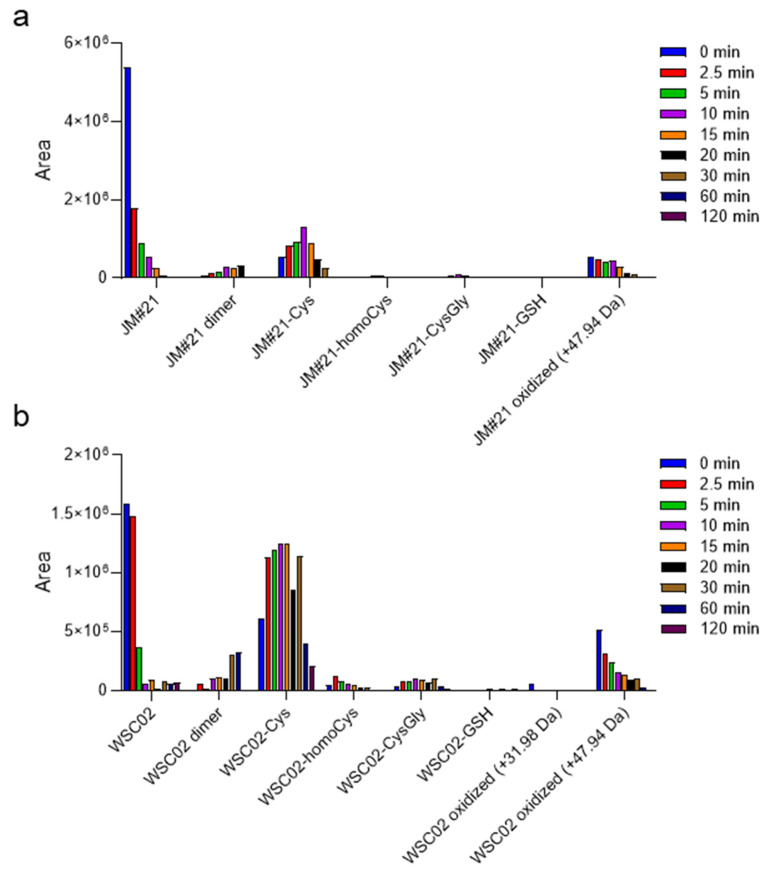
EPI-X4 JM#21 and WSC02 interact with low molecular weight compounds in human plasma via formation of thiol bonds. Aliquots of the mixture of plasma with (**a**) JM#21 or (**b**) WSC02 were taken at t = 0, 2.5, 5, 10, 15, 20, 30, 60, and 120 min and analyzed by mass spectrometry for the presence of disulfide linked cysteine (Cys, +119.0041 Da), homocysteine (homoCys, +133.0198 Da), cysteinylglycine (CysGly, +176.0255 Da), and glutathione (GSH, +305.0682 Da). In addition, their respective dimers and some oxidized (+31.98 Da, +47.94 Da) forms were investigated.

**Figure 4 ijms-23-15029-f004:**
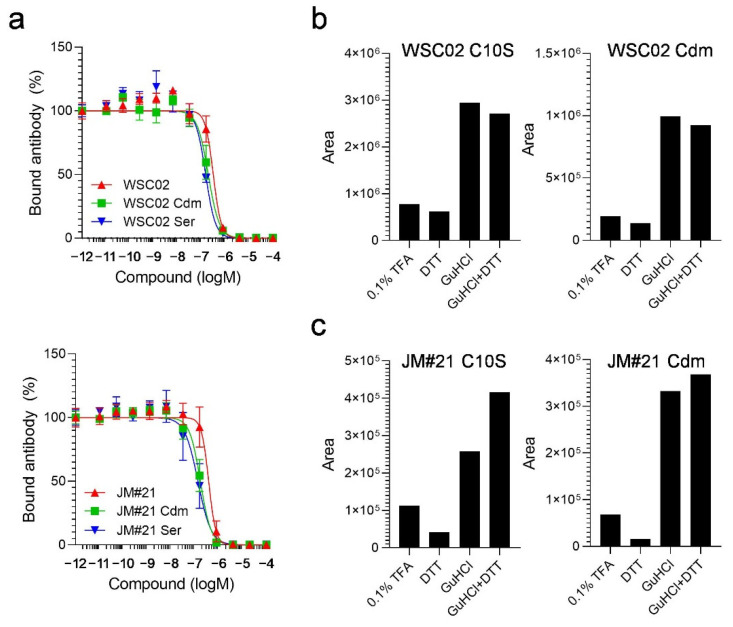
Biological activity and mass spectrometry analysis of thiol-lacking EPI-X4 WSC02 and JM#21 in plasma. Peptides with Cys replaced by Ser (C10S), or with a derivatized Cys (Cdm), are still functionally active and detectable by mass spectrometry in human plasma without previous treatment with reducing or chaotropic agents: (**a**) Peptide variants compete with the CXCR4-specific 12G5-antibody. Data from three individual experiments ± SEM are shown; (**b**,**c**) variants of WSC02 (**b**) or JM#21 (**c**) were incubated in human plasma for 10 min at 37 °C, and then treated with either 0.1% TFA, DTT, GuHCl, or a combination of GuHCl and DTT. Later, the samples were analyzed by mass spectrometry for the detection of peptide derivatives. The area values in (**b**) and (**c**) are calculated from the corresponding peak in the MS extracted ion chromatogram of every sample.

**Figure 5 ijms-23-15029-f005:**
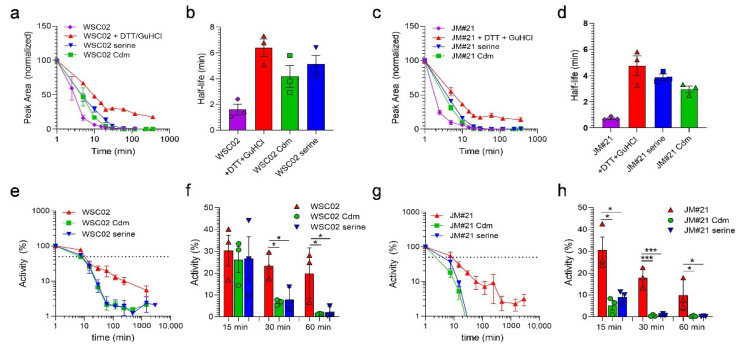
Cysteine residues in EPI-X4 WSC02 and JM#21 increase their stability in plasma: (**a**–**d**) Half-lives of WSC02, JM#21, and modified variants detected by mass spectrometry, (**a**,**c**) intact peptide normalized to t = 0 min; (**b**,**d**) half-life values determined by one-phase decay. (**e**–**h**) Stability of peptides analyzed by a functional stability assay. (**e**,**g**) Peptides were spiked into human plasma and incubated at 37 °C. Remaining activity was determined by antibody competition assay. (**f**,**h**) Remaining activity after 15 min, 30 min, and 60 min. Shown are data derived from 3 individual rounds of plasma incubation ± SEM. * *p* < 0.05, *** *p* < 0.0001 (one-way ANOVA, Tukey’s multiple comparison test).

**Figure 6 ijms-23-15029-f006:**
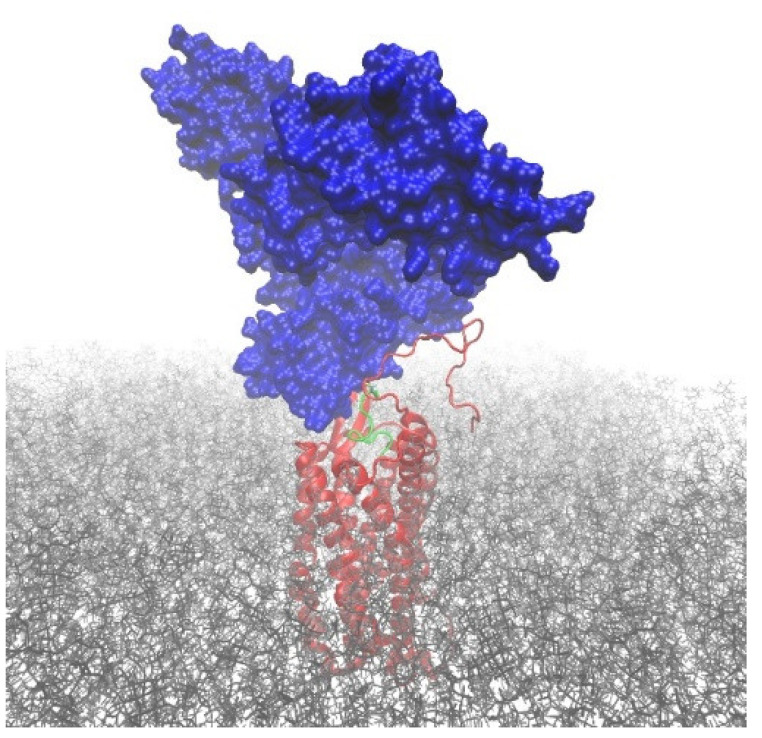
Computational modeling of EPI-X4 JM#21 (green) bound to albumin (blue) interacting with CXCR4 (red). The disulfide bond bridging albumin and JM#21 involves the residues Cys34(HSA) and Cys10(JM#21). The POPC membrane is shown in grey. Explicit water molecules and ions, omitted in the figure for clarity, were included in the model.

**Figure 7 ijms-23-15029-f007:**
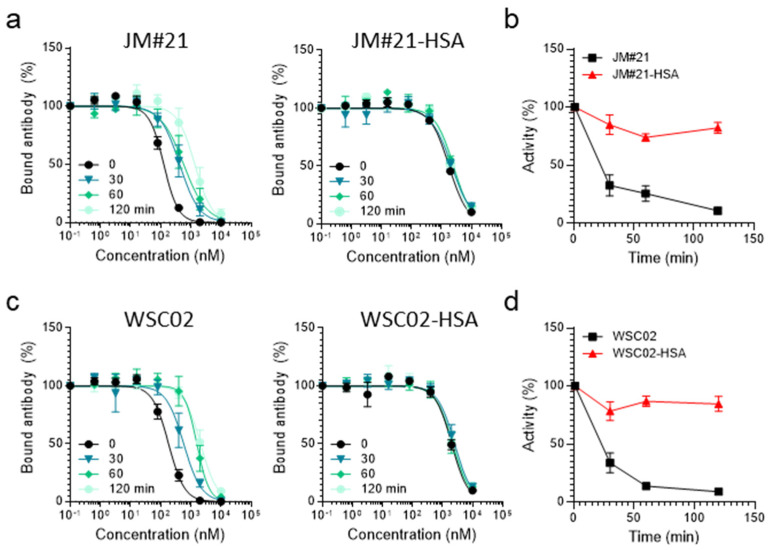
Peptides that are covalently coupled to HSA are protected against enzymatic degradation in whole human plasma as determined by functional antibody competition assay. JM#21 or JM#21 conjugates (**a**) or WSC02 or WSC02 conjugates (**c**) were spiked into human plasma and incubated at 37 °C and remaining activity analyzed in the antibody competition assay. IC_50_ values were determined by non-linear regression curve fit in GraphPad Prism. Remaining activity of JM#21 and JM#21-HSA (**b**) as well as of WSC02 and WSC02-HSA (**d**) was determined by IC_50_ (t)**/**IC_50_ (t = 0) × 100) and half-lives determined by one-phase decay. Data from at least three individual rounds of plasma incubation ± SEM are shown.

**Table 1 ijms-23-15029-t001:** Peptides used in this study.

Peptide Name	Sequence	Purity	Reference
EPI-X4 WSC02	H-IVRWSKKVP**C**VS-OH	96.5%	Zirafi et al. [7]
EPI-X4 JM#21	H-ILRWSRKLP**C**VS-OH	95.2%	Harms et al. [8]

## Data Availability

The datasets generated during and/or analyzed during the current study are available from the corresponding author on reasonable request. Mass spectrometry raw data has been uploaded to MassIVE repository, https://massive.ucsd.edu/ProteoSAFe/static/massive.jsp (accessed on 26 November 2022).

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
