# Peer review of "Advanced EPI-X4 Derivatives Covalently Bind Human Serum Albumin Resulting in Prolonged Plasma Stability"

_ijms, 2022, doi:10.3390/ijms232315029_

Round 1
Reviewer 1 Report
The manuscript provides important insights regarding details to be followed-up for stability testing of peptides. Moreover, the paper shows how to scientifically evaluate contradicting results and to verify outcomes by independent methods. In addition, binding hypotheses were confirmed by model building fitting the assumption of binding later constructs toCXCR4.
Author Response
Reviewer 1 stated “The manuscript provides important insights regarding details to be followed-up for stability testing of peptides. Moreover, the paper shows how to scientifically evaluate contradicting results and to verify outcomes by independent methods. In addition, binding hypotheses were confirmed by model building fitting the assumption of binding later constructs to CXCR4”.
Response: We thank the reviewer for his/her very positive feedback
Reviewer 2 Report
This manuscript shows an interesting effect that the conjugation of a pharmaceutically active peptide improves the half-life in plasma considerably. Unfortunately, the manuscript is difficult to read, and the introduction does not give a good understanding of the system. As a chemist, I would prefer to get an idea about CXCR4, EPI-X4, WSC02, and others. Even in the abstract, these abbreviations are used without further explanation. If some structure models should exist, they might be shown in the introduction. Also, the sequences of the respective peptides are somewhere hidden in the text "desert"; they should be shown in a figure.
Of the peptides, some specific information might be lacking, such as purity, termini, and perhaps the counterions.
Fig. 6 seems to be too small and blurry. A different representation might be more insightful.
Considering the short conclusion, which is quite clear, the results section is very long and might be condensed. Many text sections are quite lengthy and would profit from condensation.
Also, most figure legends are quite long. Most of the information might be shifted to the Materials & Methods section.
In the context of a mass spectrometric study, I would have expected some information about the degradation pathways or at least some degradation products of the respective peptides.
Most parts of the Appendices should be shifted to a supplement, except B1 and B4, which might be important for the understanding and should hence be placed in a more prominent position in the main paper.
Non-essential sections should be put in a Supplement, however without losing the context.
Perhaps the authors can obtain figures of higher resolution, e.g., for the mass spectra A1, A2, and A3. It seems that jpg formats have been used, which is definitely not a good choice for line figures. Either TIFF or vector formats are preferable for such purposes.
The scientific content of the manuscript seems to be fine. However, a complete reorganization of the structure of the paper and a condensation of the text would be very desirable.
Author Response
Reviewer 2 stated “This manuscript shows an interesting effect that the conjugation of a pharmaceutically active peptide improves the half-life in plasma considerably. Unfortunately, the manuscript is difficult to read, and the introduction does not give a good understanding of the system. As a chemist, I would prefer to get an idea about CXCR4, EPI-X4, WSC02, and others. Even in the abstract, these abbreviations are used without further explanation. If some structure models should exist, they might be shown in the introduction. Also, the sequences of the respective peptides are somewhere hidden in the text "desert"; they should be shown in a figure”.
Response: We thank the Reviewer for the evaluation of our manuscript and his/her comments. We have now added an explanation for the abbreviations in the abstract and included a new table showing the sequences of the peptides used in this study (Table 1, page 3). The introduction is written for a broad interdisciplinary audience and describes CXCR4 and its importance in the field, as well as EPI-X4 and the optimized compounds EPI-X4 WSC02 and EPI-X4 JM#21. A computational model of the complex auf CXCR4 with EPI-X4 and optimized derivatives has also previously been published, the reference is included (Sokkar et al. Bioconjug. Chem. 2022, Reference 11).
Of the peptides, some specific information might be lacking, such as purity, termini, and perhaps the counterions.
Response: We added RP-HPLC profiles and purity information about both peptides to the supplement (Figure S1 and S2) and Table 1. Also, mass spectrometry analysis to confirm that the synthetic peptide has the expected molecular mass (and therefore the complete sequence) can be found in Figure S1 and S2. Information about the counterion, trifluoroacetate (TFA), was added to the material and methods section (Page 14, Peptide synthesis, line 475).
Fig. 6 seems to be too small and blurry. A different representation might be more insightful.
Response: For the resubmission we uploaded all figures in high resolution. Figure 6 will be bigger. Also, we changed the presentation of the structure to make the figure clearer.
Considering the short conclusion, which is quite clear, the results section is very long and might be condensed. Many text sections are quite lengthy and would profit from condensation.
Response: We considered the reviewers comment and shortened some parts of the Results. However, in our view, most information is needed to understand the context of the paper. Especially for a reader who is not from the chemistry field, this information is necessary to understand the context of the paper.
Also, most figure legends are quite long. Most of the information might be shifted to the Materials & Methods section.
Response: Figure legends were shortened. All information was included in the materials and methods section.
In the context of a mass spectrometric study, I would have expected some information about the degradation pathways or at least some degradation products of the respective peptides.
Response: we thank the reviewer for this valid point. We have analyzed the exact mechanism of degradation for both peptides, however, this information will be published in a second paper together with a more advanced structure-activity-relationship study. This paper is already submitted and is available as a preprint (https://www.authorea.com/doi/full/10.22541/au.166733107.78559129/v1). In the preprint, we analyzed the mode of degradation of EPI-X4 lead compounds in human plasma using MS-based methods and found that they are exclusively degraded at the N-terminal region. This allowed us to design more advanced and more stable derivatives, which are now used in further studies.
Most parts of the Appendices should be shifted to a supplement, except B1 and B4, which might be important for the understanding and should hence be placed in a more prominent position in the main paper.
Response: Appendix A was shifted to the SI. However, we decided to keep all information of HSA-peptide conjugation combined in the Appendix to give a complete overview about the synthesis and purity control. The information about the conjugation process might be very important for a reader from the chemistry field and Appendix a will provide all information. However, for readers from other fields this information might be less relevant and interrupt the reading.
Non-essential sections should be put in a Supplement, however without losing the context.
Response: We considered the reviewers comment and moved Figure 7 to the SI. In our view, all other information is important to fully display the interaction of the peptides with plasma components, such as HSA and other low molecular weight compounds.
Perhaps the authors can obtain figures of higher resolution, e.g., for the mass spectra A1, A2, and A3. It seems that jpg formats have been used, which is definitely not a good choice for line figures. Either TIFF or vector formats are preferable for such purposes.
Response: For resubmission all figures were uploaded in high resolution.
The scientific content of the manuscript seems to be fine. However, a complete reorganization of the structure of the paper and a condensation of the text would be very desirable.
Response: We feel that with the changes introduced and considering the very positive evaluation by reviewer 1, our manuscript is now well readable by the interested reader.
Round 2
Reviewer 2 Report
The authors improved the manuscript considerably. However, it is still very long and not really reorganized. Nevertheless, it might be publishable now.
